# Ras Family of Small GTPases in CRC: New Perspectives for Overcoming Drug Resistance

**DOI:** 10.3390/cancers13153757

**Published:** 2021-07-26

**Authors:** Anxo Rio-Vilariño, Laura del Puerto-Nevado, Jesús García-Foncillas, Arancha Cebrián

**Affiliations:** Translational Oncology Division, Hospital Universitario Fundación Jimenez Diaz, 28040 Madrid, Spain; anxo.rio@quironsalud.es (A.R.-V.); lpuerto@oncohealth.eu (L.d.P.-N.)

**Keywords:** colorectal cancer, drug resistance, EGFR targeted therapies, Ras-GTPases

## Abstract

**Simple Summary:**

Ras-GTPases play a key role in the control of fundamental cellular processes such as proliferation, survival, adhesion, and differentiation. These functions make them particularly relevant in the development and progression of numerous types of cancer. Activating mutations in some of these proteins are particularly relevant in colorectal cancer and largely determine its response to available therapies. In this review, we will discuss the role of Ras-GTPases in colorectal cancer, the strategies available to inhibit them and their implication in overcoming resistance to the therapies currently used in clinical practice.

**Abstract:**

Colorectal cancer remains among the cancers with the highest incidence, prevalence, and mortality worldwide. Although the development of targeted therapies against the EGFR and VEGFR membrane receptors has considerably improved survival in these patients, the appearance of resistance means that their success is still limited. Overactivation of several members of the Ras-GTPase family is one of the main actors in both tumour progression and the lack of response to cytotoxic and targeted therapies. This fact has led many resources to be devoted over the last decades to the development of targeted therapies against these proteins. However, they have not been as successful as expected in their move to the clinic so far. In this review, we will analyse the role of these Ras-GTPases in the emergence and development of colorectal cancer and their relationship with resistance to targeted therapies, as well as the status and new advances in the design of targeted therapies against these proteins and their possible clinical implications.

## 1. Introduction

According to the Global Cancer Observatory (GLOBOCAN) reports, colorectal cancer (CRC) ranks third in the number of diagnoses, representing 10% of the total, and being the second one in prevalence and mortality. This means more than 1.4 million new cases and near 1 million deaths worldwide in 2020 [1]. However, screening and early diagnosis programs are increasing the detection of tumours at earlier stages, significantly increasing the overall survival of patients [2].

### 1.1. Colorectal Carcinogenesis

More than 95% of all CRCs diagnosed worldwide occur spontaneously, while the remaining small proportion has hereditary components associated with certain syndromes such as Lynch, Muir-Torre or Turcot syndromes [3,4]. Within sporadic cancers, most (up to 85%) begin with the formation of a benign adenoma [5], because of the accumulation of changes related to chromosomal instability (CIN) or microsatellite instability (MSI) [6]. CIN events involve an increased propensity for errors in chromosome segregation during mitosis, leading to the appearance of aneuploidies that is characteristic of many cancers [7], as a result of the sequential acquisition of genetic and epigenetic changes [6]. Thus, bi-allelic mutations in *APC* (adenomatous polyposis coli) lead to the formation of an early adenoma, which progresses to a late adenoma through activating *KRAS* mutations and loss of chromosome 18q, which leads to a loss of *SMAD4*. Finally, deletion of chromosome 17p leads to the loss of *TP53*, promoting the malignant transformation of late adenoma to a colorectal carcinoma [6], due to the dysregulation of Wnt/β-catenin, MAPK, PI3K, and TGF-β pathways. On the other hand, microsatellite instability (MSI) is characterized by defects in the DNA mismatch repair system that led to initial alterations of Wnt pathway associated with the development of an early adenoma. Meanwhile, *BRAF* mutations, together with alterations of *TGFBR*, *IGF2R*, and *BAX*, promote tumour progression toward intermediate and late states of carcinogenesis [8].

In addition, there is an alternative pathway that leads to sporadic CRC, known as the serrated pathway, which accounts for about 15–40% of all diagnoses [9]. This pathway begins with the formation of a hyperplastic polyp, followed by the formation of a sessile serrated adenoma and, finally, serrated CRC [5]. These tumours have a CpG Island Methylator Phenotype (CIMP) characterized by the hypermethylation of tumour suppressor gene promoters, like *EPHB2*, leading to overactivation of MAPK pathway, and a high level of MSI [10,11,12].

### 1.2. Standard of Care for Managing CRC Patients. Status of Targeted Therapies

The standard of care treatment for CRC patients is associated with the possibility of surgical resection of the tumour. In non-metastatic patients, this resection is always possible, and, depending on the tumour stage, it may be or not accompanied by other treatments. For patients with Stages 0 and I, surgery is enough by itself as treatment, being no evidence about an increase of overall survival when adjuvant therapy is given. On the other hand, patients with Stages II and III CRC often receive radiotherapy and conventional adjuvant chemotherapy, respectively [13]. Within the latter, irinotecan, 5-FU (5-fluuoruracil), and oxaliplatin stand out as first-line treatments [14]. However, metastatic CRC (mCRC, Stage IV) patients do not usually undergo surgery. Most of them show liver metastasis, and resection is only possible in 15% of them [15]. In all these patients, targeted therapies, like bevacizumab and cetuximab, are widely used in combination with conventional chemotherapy like FOLFOX (5-FU, leucovorin, and oxaliplatin) or FOLFIRI (5-FU, leucovorin, and irinotecan) either as neoadjuvant therapy in patients who are candidates for surgery or as palliative therapy in patients for whom surgery is not possible [16,17]. The use of targeted therapies to manage mCRC patients has increased the overall survival from 18.8 to more than 30 months [14].

The clinical and molecular heterogeneity of CRC [18] makes the patient’s outcome and response to available therapies highly variable [19]. For this reason, in recent years, many attempts have been made to establish a classification based on molecular characteristics to thus improve the clinical accuracy of treatments [20]. Currently, the most widely accepted molecular classification systems for CRC are The Cancer Genome Atlas (TCGA) and the Consensus Molecular Subtypes (CMS) [19] (Figure 1). Nevertheless, due to the great complexity to determine these subtypes in clinical practice, their application is still limited. It is, therefore, necessary to determine a much simpler classification that represents the main tumour types based on their molecular status to improve the design of new treatments and the effectiveness of the existing ones. Thus, there is a growing need for more sophisticated targeted therapies to address the problems associated with conventional chemotherapy, which is currently the most used as first-line treatment [21] and has major associated issues such as low specificity, high systemic toxicity, and the unpredictable emergence of acquired resistance [22].

Currently, different targeted therapies for CRC based on small molecules or monoclonal antibodies directed to specific proteins are available [23]. Bevacizumab was the first approved by the Food and Drug Administration (FDA). It consists of a monoclonal antibody that is specifically targeted against VEGF-A, preventing its binding to their receptors, VEGFR 1–2, inhibiting the angiogenic and pro-survival response mediated by this pathway and reducing the tumour growth [24,25]. At this moment, other VEGF signalling pathway inhibitors, like ziv-aflibercept, regorafenib, or ramucirumab are allowed by the FDA [21].

The second targeted therapy approved for treating CRC is based on the inhibition of EGFR-mediated signalling. The two main drugs approved with this proposal were Cetuximab and Panitumumab. Both are monoclonal antibodies directed to EGFR, whose pathway has been related to pro-angiogenic activities, cell proliferation, cell apoptosis, migration, and invasion [26]. However, only a small number of patients, near 20%, can benefit from these therapies due to primary and acquired resistance [27]. Although many of the mechanisms of cetuximab resistance have not been described yet, it is well documented that the overactivation of Ras-GTPases, especially KRAS and NRAS, both downstream of EGFR, play an essential role in this process [28]. Therefore, it is essential to understand the biology of Ras-GTPases, and their implications for tumour progression and drug response not only to ensure the correct choice of treatments but also for the design of combined therapies to improve their efficacy.

In recent years, a plethora of Ras-GTPase inhibitors have been emerging with promising results in preclinical research, some of which are already under clinical trials [29]. Thus, it is expected to be a turning point in the treatment of *KRAS*-mutated CRC and in overcoming resistance to anti-EGFR therapies. In the following paragraphs, we will discuss the status of Ras-GTPases, their involvement in CRC progression and their role in the response to therapies, as well as an update on the new inhibitors that are emerging and their possible implications.

## 2. Ras-GTPase Family

Ras-GTPases represent one of the five main families in which Small-GTPases are divided. The Ras family includes 36 proteins arranged in 6 subfamilies (reviewed in Table 1). Together, all the small GTPases share the ability of binding to GTP, inducing a conformational change that makes them transit to an active state, which allows them to interact with other proteins, triggering the activation of different signalling pathways. On the other hand, when binding to GDP, these proteins reach an inactive conformation, permitting them to act as a “*molecular switch*” [29,30,31] that can control many cellular processes such as vesicle trafficking, molecular transport through the nucleus, cell shape maintaining and polarity, or cell cycle control and proliferation. In this context, Ras is the largest and most diverse of all the families that are comprised by the Small GTPases and play an important role in the control of the cell cycle and proliferation, which makes it particularly relevant in the appearance and progression of almost all types of cancer [32].

Structurally, the Ras family consists of proteins with lengths of 160–180 amino acids [60], with a highly conserved G domain in the N-term end, in which resides the catalytic activity, and a hypervariable region (HVR) in C-term end which varies among the different members of the family, and which will largely determine the post-translational modifications of these proteins (Figure 2) [32,60]. Thus, the G domain consists of five highly conserved regions named from G1 to G5. G1 corresponds with a P-loop, in which resides the catalytic activity of the protein. Conversely, G2 and G3 determine the activity of the protein, reaching different conformations depending on whether they are bound to GTP or GDP, thus determining their interaction with other proteins [61]. The hypervariable region, in most cases, contains a CAAX motif, with C being a cysteine, A an aliphatic residue, and X any amino acid. This motif is a post-translational target for prenylation, which consists of the covalent addition of isoprenyl group (farnesyl, 15C, or geranylgeranyl, 20C) to ensure the subcellular location of Ras members in cell membranes, which is essential for its signalling activity [62,63] (Figure 2). Additionally, many Ras-GTPases require additional structural modifications to localize specifically to the plasma membrane. In some cases, as occurs in NRAS and HRAS, it is achieved by the addition of palmitoyl to cysteine residues present in the hypervariable domain. In other members like KRAS, this is solved by the presence of a polybasic domain that allows electrostatic interaction with the plasma membrane, which is negatively charged [64] (Figure 2). However, the exception to this pattern is found in the RGK subfamily, which lacks the CAAX domain and therefore does not present these post-translational modifications, showing a mainly cytosolic localization [34].

### 2.1. Regulation of Ras-GTPase Activity

#### 2.1.1. Regulating GTP/GDP Exchange

As previously commented, all the members of Ras-GTPase family have intrinsic GTPase activity which allows them to hydrolyse GTP to GDP. However, this reaction occurs at low rates by itself and requires the participation of accessory molecules to catalyse it. GTPase-Activating Proteins (GAPs) and GDP Exchange Factors (GEFs) are the two main proteins involved in regulating the function of Ras-GTPases and both coexist in cells improving the effectiveness of signalling regulation [65,66,67] (Figure 3). Thus, GAPs promote the hydrolysis of GTP to GDP. Once hydrolysis has occurred, GEFs destabilize the binding between GTPase and GDP, promoting its dissociation and allowing GTP to bind again reaching the active conformation. The preferential binding of GTP to catalytic region is not caused by differences in affinity but occurs as a consequence of much higher concentrations of GTP compared to GDP [61].

#### 2.1.2. Regulating Subcellular Location

Additionally, post-translational modifications display essential functions in Ras-GTPase regulation. Prenylation is the most important of these modifications. It consists of the binding of a farnesyl or geranylgeranyl group to the CAAX motif. Farnesyl groups bind preferentially when X residue is methionine, serine, glutamine, or alanine, while geranylgeranyl groups have a higher affinity for CAAX motifs harbouring leucine or phenylalanine. Following prenylation, the CAAX motif is cleaved by Rce1 and C-terminal end is methylated by Protein-S-isoprenylcysteine O-methyltransferase (ICMT) [68,69]. It is essential to determine the correct subcellular localization of these proteins, the protection against proteasomal degradation, and the protein–protein interactions [70].

Apart from prenylation, there are other important post-translational modifications in the regulation of subcellular locations of Ras-GTPases. For example, palmitoylation has a relevant effect on NRAS and HRAS proteins. It is required for their trafficking from the endomembrane system to the plasma membrane [63], increasing the affinity generated by prenylation to bind to the plasma membrane. This reversible mechanism allows regulation of Ras activity by modifying its relative amount through the endo-membrane system [71,72]. Thus, palmitoylation acts directly on the activity of NRAS and HRAS, since their dissociation from the membrane causes a decrease in the intensity of their signalling pathways [73]. This modification is also essential for allowing their location at the plasma membrane or ensuring their attachment to endosome membranes [74] of other non-classical Ras proteins as Ral1B [75] or ERAS42 [76]. On the other hand, there are other members such as KRAS that are not modified by palmitoylation. Instead, they have a polybasic lysine tail that endows them with a negative charge and leads them to establish electrostatic interactions with the negatively charged plasma membrane [77].

Membrane binding may also be modulated by phosphorylation, which neutralizes the positive charges and reduces their affinity for the plasma membrane, favouring subcellular localization in endosomes [78]. This phenomenon has been fully described in certain members of Ras family as KRAS4B, whose Serine 181 is a substrate for Protein Kinase C (PKC) and cGMP-dependent Protein Kinase 2 (PKG2), regulating the association of its HVR with membrane microdomains [79]. It has also been reported for the RAL subfamily that it is a target of Aurora Kinase A (AURKA), a mitotic serin-threonine kinase, which phosphorylates S194 and S198 residues of RALA [80] and RALB [38], respectively. Again, this phosphorylation regulates the subcellular location of RAL proteins, leading to relocation from plasma membrane to the endomembrane system and allowing the interaction with their main effector RALBP1 to mediate anchorage-independent growth and tumorigenesis [80]. In the case of RHEB, the role of phosphorylation has not yet been related to its subcellular location, although this modification reduces its activity and leads to a loss of signalling of the downstream mTOR pathway [81].

## 3. The Role of Ras-GTPases on CRC

The RAS subfamily is the best characterized one in carcinogenesis, playing a major role in the cell cycle control, proliferation, adhesion, differentiation, and migration. It has led to most of its members being considered oncogenes, acquiring particular importance in CRC since the first stages of tumour development. Thus, nearly 45% of CRC harbour activating mutations in *KRAS*, *HRAS* or *NRAS* [79,82], most of them affecting codons 12 and 13 in *KRAS* and *HRAS*, and codon 61 in *NRAS* [83]. The constitutively active state of these proteins promotes RAS binding to BRAF leading to overactivation of several signalling pathways implicated in cell proliferation, including PI3K/AKT, MEK, and ERK pathways [49,84] and, more recently documented, Wnt/β-catenin pathway [85]. Additionally, during the last lustrum, special attention has been focused on the importance of oncogenic RAS in metabolic modulation (exhaustively reviewed by Mukhopadhyay et al. in 2021) [86]. It has been described as a hallmark of cancer and plays a critical role y *KRAS*-mutated CRC [83], as will be described in later sections.

Among the RAS subfamily, DIRASs have recently emerged as a tumour suppressor Ras-GTPases. It has been shown that DIRASs modulate JAK/SAT, PI3K/AKT, mTOR, and NF-κβ pathways, which have been demonstrated to be essential in the control of cell proliferation [50]. DIRAS3 downregulation has been reported in ovarian and breast cancer compared with healthy tissues and its low expression was also associated with tumour invasiveness and higher aggressiveness [87]. In many cases, silencing and down-regulation of these proteins are caused by hypermethylation events, which are frequent in CRC (as commented in the introduction). Thus, regarding its role in CRC, it has been shown that aberrant methylation of *DIRAS1* is related to the poor prognosis of CRC patients, and it acts as an inhibitor of cell proliferation, migration, and invasion in CRC. Finally, overexpression of DIRAS in tumour xenograft mice led to a slowdown in tumour growth [88].

Moreover, up to 9 additional RAS effectors related to cell proliferation have been documented [89]. However, they must be better characterized, including mutation screening, to improve the clinical management of cancer patients. One of these effectors is the RalGEF-Ral signalling pathway, which involves the RAL subfamily. In recent years, both RALA and RALB have been reported as important downstream of KRAS [37]. They seem to play a crucial role in the development and progression of different types of tumours. RALA protein was related to cell proliferation in some *KRAS*-mutated cells, especially those harbouring G12C or G12V mutations [90]. Additionally, both in vitro and in vivo approaches identify this protein with the development of metastasis through the action of its effector RALBP1. Thus, silencing both RALA and RALB leads to a drop in their invasiveness and metastatic capacity in NSCLC [91]. However, the scientific evidence regarding the role of RALB in cancer is controversial. Many reports associate it with the survival and metastatic capacity of human tumour cells [39,92]. However, its expression has also been related to a decrease in cell proliferation, acting, in this case, as a tumour suppressor. Particularly in CRC, overexpression of RALA has been associated with an increase of cell proliferation and shorter relapse-free survival whereas, RALB depletion has been related both with the increasing of colony-formation capacity [93], but also with increased apoptosis, so its role in CRC remains unclear [94]. So far, few studies have been performed to determine the role of RALA and RALB in CRC and their interaction networks. Therefore, further research is necessary for a better understanding.

Regarding the RHEB subfamily, although it does not interact with the oncogenic members of the classic RAS subfamily, it is a major component of the mTOR pathway [57]. RHEB is essential in the regulation of cell growth, proliferation, response to growth factors and, more recently, angiogenic processes. This signalling pathway is dysregulated in many types of tumours [95], usually as a consequence of mutations upstream of mTOR, including activating mutations in *PI3K* (present in 32% of CRC), and generally associated with late stages of tumorigenesis [96]. Inactivating mutations in *PTEN*, a negative regulator of PI3K that activates the mTOR signalling pathway, has been shown to increase the oncogenic processes [97]. Additionally, RHEB is highly expressed in CRC tissues compared with normal tissues, and its silencing inhibits the activation of the mTOR signalling pathway reducing cell proliferation and promoting apoptosis [55,56].

Finally, the RAP subfamily is involved in regulating processes involved in tumour cell migration, invasion, and metastasis [43]. Among the RAP effectors is AF-6, which regulates cell adhesion by interaction with p120 catenin and inhibition of E-cadherin endocytosis. Additionally, RAP1A and RAP1B play different roles in cancer, promoting translocation of the Rap Associated with DIL Domain (RADIL) from the cytoplasm to the plasma membrane, increasing cell adhesion [98]. It also interacts with TIAM1 (TIAM Rac1 associated GEF 1), and CAC2 (Chromatin assembly factor 1 subunit p60), activating RAC and CDC42 to regulate cell polarization and movement [99,100]. Regarding its role in tumour metastasis and invasive capacity, in vivo and in vitro studies point towards a pro-metastatic activity. Thus, RAP has been shown to promote metastasis in melanoma [101,102], breast cancer [42], Head and Neck Squamous Cell Carcinoma (HNSCC) [103] and pancreatic cancer [104]. Particularly in CRC, RAP1A KO inhibited cell growth through PI3K, FOXO3, and CycD [44]. Additionally, downregulation of SIPA1, a Rap1GAP, was associated with an increase in metastatic ability [105].

Taken together, these data support the importance of the Ras-GTPase family, beyond the classical oncogenic branch, in the onset and progression of CRC. Thus, further research into the other members of this family is needed, which will allow not only a better understanding of CRC but also an improvement of currently available therapies.

### 3.1. Status of Ras-Targeted Therapy

#### 3.1.1. Targeting Ras-GTPase Location

The difficulty associated with the development of specific inhibitors against oncogenic members of the RAS family, together with a better understanding of the biology of this family of proteins, has led to the design of alternative approaches that have allowed more effective strategies (Figure 4). One of the first alternatives was the development of inhibitors of Ras post-translational, essential for their subcellular location and, therefore, their signalling functions [106].

The development of Farnesyl Transferase Inhibitors (FTIs), designed for blocking Ras-GTPases location in the plasma membrane, was one of the first Ras-targeted therapies [83,107]. Among these inhibitors are Tipifarnib [108] and Lonafarnib [109], which had success in preclinical studies but failed in the translation to clinical trials. The same fate has befallen other approaches to indirectly inhibit the farnesylation of RASs, such as inhibition of PKC. PKC has been shown to alter KRAS location by phosphorylating the polybasic region. It modulates the electrostatic interaction of the farnesyl group with the plasma membrane. Thus, PKC inhibition compromised KRAS location in the plasma membrane, leading to reduced colorectal cell proliferation [110]. Unfortunately, Bryostatin, a PKC inhibitor approved by FDA for treating other pathologies, was not effective in phase 2 clinical trials in mCRC patients with mutated *KRAS* [111].

One of the reasons associated with the low success of these inhibitors in clinical trials is that some Ras-GTPases are regulated by alternative prenylations, like geranyl-geranylations, bypassing the inhibition of the farnesylation [83]. Thus, when FTIs are used, Ras-GTPases find an alternative way to maintain their localization in the plasma membrane. Since dual silencing of FTs and GGTs has not shown deleterious effects in mice, combinatory therapy of FTIs and GGTIs or Dual Prenylation Inhibitors (DPIs) have been proposed for treating KRAS-mutated tumours. So far, the most promising DPI is L-778, which has been demonstrated to be effective in leukemic cells [112]. However, the observed side effects due to the simultaneous inhibition of FTIs and GGTIs could make their clinical application unfeasible.

Currently, the efforts are focused on the search of alternative targets involved in the correct membrane location of the Ras subfamily. One of them is the ICMT methyltransferase, implicated on the Ras-GTPases C-end methylation and essential for its location. The first-generation inhibitor of ICMT was Cysmethinil, which was successful in CRC preclinical assays, reducing cell proliferation while ICMT overexpression reverted it [113]. The second generation of ICMT inhibitors, called 8.12, favour Ras relocation away from the plasma membrane, cell cycle arrest, autophagy, and cell death both in vitro and in vivo models [114]. However, further clinical trials are needed to confirm their efficacy in CRC patients.

Another approach to avoid the correct membrane location of Ras-GTPase members is the inhibition of Phosphodiesterase δ (PDE-δ), an enzyme which binds to the farnesyl end of the oncogenic branch of members of RAS subfamily, promoting its recycling through the endomembrane system [115]. Deltarasine has emerged as the most promising alternative impairing the KRAS location to endomembrane and, consequently, impairing tumour growth in CRC models with oncogenic KRAS [116]. Recently, new inhibitors against PDE-δ (Deltazinone 1) with higher specificity and less cytotoxic have been developed [117], but they have not been tested in vivo yet. Therefore, further studies are necessary to consolidate them as an alternative for clinical practice.

In addition to inhibitors against farnesylation, molecules that compete with farnesylated Ras for the plasma membrane binding site are being designed. One of them is Salirasib, a novel oral inhibitor that has shown an inhibitory effect over KRAS-mutated CRC cell lines. A synergistic effect was observed when this inhibitor was combined with Wnt pathway inhibitors [118]. In vitro studies suggest that this inhibitor partially reduces cell proliferation through inhibition of the mTOR, causing similar effects as Rapamycin [119]. Therefore, it would not be surprising that this effect could be caused by RHEB inhibition, although it has not yet been addressed. So far, it has shown good results in terms of toxicity, making it through several phase 1 trials, although its effectiveness in patients with *KRAS* mutations is still unclear [120].

#### 3.1.2. Direct Targeting of Ras-GTPases

Recently, there has been a paradigm shift in the direct inhibition of the RAS subfamily of GTPases. Although it was initially thought that this was not possible, in recent years, specific inhibitors that bind directly to these proteins have been beginning to emerge, most of them targeting KRAS, since it is one of the most frequently mutant protein in human neoplasia.

One of the main problems in developing therapies that directly target Ras is the unspecific toxicity associated with the inhibition of both wild-type and mutant isoforms. Therefore, one of the greatest milestones has been the achievement of mutant-specific inhibitors. Currently, most of those have been designed against *KRAS* G12C mutation, present in above 4% of CRC patients [48]. Thus, Oestern and colleagues demonstrated that the change of glycine to cysteine creates a new binding pocket, called binding-switch II region for which specific inhibitors could be developed avoiding the binding to WT isoforms [121]. In the same article, they provided a series of inhibitors targeting this mutant. ARS-1620 emerged as the first and proved to be highly selective and well-tolerated both in vitro and in vivo, binding at least 75% of the mutant molecules and achieving a potent anti-tumour effect [122]. In recent years, numerous KRAS-G12C inhibitors have been designed, four of them are under evaluation in clinical trials. The first, Sotorasib (AMG-510), showed the ability to maintain stable disease in patients with this *KRAS* mutation in Phase 1 trial [123]. Additionally, other two clinical trials are currently underway for non-small cell lung carcinoma (NSCLC), in phases II (NCT03600883) and III (NCT04303780). In 2020, the inhibitors MRTX849 (Adagrasib), LY3499446 and JNJ-74699157 entered clinical trials, although LY3499446 was recently discontinued due to unexpected adverse events. MRTX849 has been shown to be a safe drug in the Phase 1 KRYSTAL-1 clinical trial. At present it is in a Phase 1/2 Study to evaluate its safety, tolerability, and clinical activity in patients with cancer having a *KRAS* G12C mutation (https://clinicaltrials.gov/ct2/show/NCT03785249, accessed on 24 June 2021); and in a Phase 3 clinical trial to compare its efficacy when combined with cetuximab versus chemotherapy in patients with CRC with *KRAS* G12C mutation (https://www.clinicaltrials.gov/ct2/show/NCT04793958 accessed on 16 July 2021). Moreover, KS-58 is the first KRAS-G12D inhibitor and, so far, it has been tested in preclinical trials in lung and pancreatic cancer, showing promising results. This study demonstrated that its combination with gemcitabine, reduced tumour size by 66% [124]. However, recent evidence in CRC point that these inhibitors have a high propensity to develop acquired resistance due to reactivating feedback from native forms of KRAS [125].

Additionally, inhibitors targeting other members of the Ras-GTPase family have been developed. For example, a specific inhibitor of RALA and RALB was effective in reducing both cell proliferation and tumour growth in lung cancer xenografts [126]. Nevertheless, given the opposing roles described for RALA and RALB in CRC, further studies will be necessary before these inhibitors can be considered as a real alternative for the treatment of CRC. Additionally, a potent RHEB inhibitor, called NR1, has also been designed and seems to prevent the phosphorylation of mTORC1 in both in vivo and in vitro assays. Thus, RHEB inhibitors appear as an alternative avenue for the development of new therapies targeting the mTOR pathway that should be investigated.

#### 3.1.3. Targeting Ras Upstream Effectors

Failure of the first Ras-GTPases inhibition approaches which aimed to impede their correct location in the cell membranes along with the late development of RAS-selective inhibitors has led to the search for alternative procedures. Between these strategies are those focused on the inhibition of GDP/GTP Exchanger Proteins, especially those linked to the oncogenic members of Ras subfamily. Inhibition between KRAS and Son of Sevenless (SOS), a GEF that promotes GDP/GTP exchange leading to KRAS activation, has been the main objective of these strategies [127]. To date, many inhibitors have been developed to avoid the interaction between SOS1 and KRAS. They are designed against the catalytic site of SOS1, avoiding its dimerization with KRAS and maintaining KRAS GDP-bounded state [128]. In the case of CRC, the best characterized is BI-1701963, which is under clinical evaluation for metastatic CRC patients with mutant KRAS (https://clinicaltrials.gov/ct2/show/NCT04627142 accessed on 10 May 2021). Additionally, BI-3406 has arisen as a promising alternative showing high specificity in inhibiting CRC cell growth harbouring *KRAS* mutations, especially in combination therapy with MEK inhibitors [129].

In the case of the RAL-GTPases subfamily, it has been described that its function is partially regulated through its phosphorylation in C-Terminal end S194 by AURKA, favouring its activation and translocation from the plasma membrane to endomembrane system, necessary for KRAS-mediated oncogenic transformation [80]. However, the opposite results found about the regulation of RALA by AURKA make further study necessary before deciding whether its inhibition is a good approach to indirectly target Ras-GTPases. Although it has been shown that AURKA targeted therapies lead to a downstream inhibition of RALA, regulating anchorage-independent cell growth [130], other studies show that the effect of AURKA inhibition does not affect RALA function [89]. Since AURKA overexpression has been related to chromosomal instability and deregulation of Wnt and RAS pathways [131], it would not be surprising that RALA GTPases, downstream of AURKA, could have a key role in CRC. Defining the role of RALA in CRC and its relationship with AURKA would be interesting for improving the selection of patients who could benefit from anti-AURKA therapies that are currently in clinical trials, with Alisertib (MLN8237) being the most promising of its inhibitors.

#### 3.1.4. Targeting Ras Downstream Effectors

Another interesting approach with a long clinical track record is found in the inhibition of downstream molecules of Ras-GTPases. This strategy has been widely studied and is essentially focused on MAPK and PI3K signalling pathways. BRAF kinase is the first effector of the MAPK signalling pathway and is immediately downstream of RAS and, when active, phosphorylates MEK triggering its activation [132]. Under this premise, several inhibitors have been designed against BRAF, or BRAF/MEK function, such as vemurafenib, dabrafenib or trametinib, but all of them failed to improve progression-free survival or overall survival in patients with mutant *KRAS*. Inhibition of BRAF is thought to generate a compensatory effect by the overactivation of MAPK [21] and PI3K [133] signalling pathways. Indeed, preclinical research has shown that combinatory therapies inhibiting BRAF and upstream pathway effectors would be more effective than the single inhibition of BRAF [133,134,135]. However, so far there are no ongoing clinical trials to evaluate these combinations.

Regarding MEK inhibition, many inhibitors like Binimetinib, Cobimetinib, or Trametinib have been approved by the FDA for other indications. However, they also failed when used as monotherapy in clinical trials for melanoma, NSCLC, and CRC with *RAS* or *RAF* mutations [136]. Although combinatory therapies using these inhibitors have not yet been evaluated in patients who carry *RAS* mutations, they are being assessed in patients harbouring *BRAF* mutations, but no results are available at this time. On the other hand, inhibition of ERK, the last kinase of the cascade, has been aroused as a promising alternative for treating RAS overactivation. Compounds like the preclinical SCH-772984 or its improved version MK-8353 bind to ERK1/2 promoting its inactivation by avoiding its phosphorylation by MEK [137,138,139]. Both inhibitors decrease cell proliferation in preclinical assays, but did not generate antitumour responses in clinical trials [138]. The same occurred with GDC-0994, a selective ATP-competitive inhibitor of ERK1/2. In combination with cobimetinib (MEK inhibitor), it significantly reduces tumour growth in xenograft models, while when it was evaluated as monotherapy these effects were not achieved [109].

As mentioned above, inhibition of the PI3K/AKT signalling pathway has been also proposed as an interesting approach for treating *RAS*-mutated cancers. Although there are available FDA-approved drugs for targeting PI3K, being Alpelisib the best characterized, none of them are approved for its use in *RAS*-mutated cancer patients [140]. This strategy has more limitations because, while mutations in *RAS* and MAPK are exclusive, both can coexist with *PI3K* mutations [48]. This implies that *KRAS* mutations are enough per se to generate deregulation in MAPK but not in PI3K. Additionally, inhibition of MAPK signalling could promote a compensatory overactivation of PI3K [141], so *RAS*-mutated patients could be benefited from combinatory therapies [140]. The theoretically promising approach of combination therapy with MAPK and PI3K inhibitors was toxic in clinical trials [142,143]. Alternatively, the inhibition of IGF1R, which is involved in PI3K activation in *RAS*-mutated cells, and MAPK has been successfully tested in CRC preclinical trials [144]. Likewise, it was observed that replacing MAPK inhibitor with the covalent KRAS-G12C inhibitor ARS-1620 (reviewed in the previous sections) in these combinations improved efficacy and tolerability in murine models [145]. Now, the toxicity and efficacy of these combinations need to be evaluated in clinical trials.

Finally, since both RALA and RHEB have been involved in Ras-mediated oncogenic transformation, direct inhibition of these proteins could be an alternative therapy, single or in combination, to treat tumours with oncogenic RAS mutations.

#### 3.1.5. Targeting Ras-Mediated Metabolic Reprogramming

Metabolic reprogramming was mentioned in the previous sections as a frequent phenomenon in many KRAS-mutated cancers [146]. Cancers with metabolic reprogramming usually show dependence on aerobic glycolysis, glutaminolysis and it is accompanied by increased nutrient uptake. Additionally, the biosynthesis of fatty acids and precursors of nucleic acids and glycosylation are frequent [86]. Thus, this enrichment of anabolic metabolism allows these cells to sustain their uncontrolled proliferation. However, it also reveals some metabolic vulnerabilities [147], leading to the emergence of new ways of approaching the treatment of patients with *KRAS* mutations through their metabolic modulation.

The first of the strategies to address the treatment of *KRAS*-mutated tumours through metabolic modulation consists of interfering with glucose metabolism. This topic has been extensively reviewed since Otto Warburg discovered that cultured tumour cells had an enrichment of glycolysis rates even in presence of oxygen (Warburg’s Effect) [148,149]. In this context, Vitamin C has emerged as an unexpected but promising alternative. It has been shown that Vitamin C inhibits GAPDH and impairs cell growth specifically in *KRAS* and *BRAF*-mutant CRC cell line by interfering with glycolysis [150]. Aguilera et al. reported that Vitamin C is also implicated in KRAS uncoupling from the plasma membrane, disrupting the expression of key metabolic enzymes and promoting cell death in KRAS-mutated cells [151]. Recently, the same group demonstrated that this compound, at pharmacological doses, can also modulate Krebs Cycle through PDK-1 (pyruvate dehydrogenase kinase 1) inhibition, again killing cells harbouring *KRAS* mutations but not those that are wild type [152]. Currently, Vitamin C is under clinical trials (see Table 2), one of which is already in phase III, studying its effect in combination with FOLFOX or Bevacizumab (https://clinicaltrials.gov/ct2/show/results/NCT02969681 accessed on 15 July 2021).

Targeting amino acid metabolism has emerged in recent years as a good strategy in several RAS-driven cancers like pancreatic ductal adenocarcinoma (PDAC) [153], lung cancer [154], or melanoma [155]. However, its role in CRC seems not to be as clear. There are some publications about the important role of GLUD1 (glutamine dehydrogenase 1) and SLC25A13 (Solute Carrier 25A13, an aspartate-glutamate mitochondrial carrier) for *KRAS* mutant CRC survival under glucose-deprivation conditions [156]. Additionally, it has been shown that ASCT2 (alanine-serine-cysteine transporter 2), an essential glutamine transporter, is upregulated by mutant KRAS, and its knockout reduces cell proliferation and migration in CRC. However, the effect of KO was much higher than glutamine deprivation so it may have other functions beyond glutamine transport [157]. Indeed, Lu et al. demonstrated in 2016 that ASCT2 is associated with EGFR [158], supporting a direct relationship with the RAS signalling cascade. Furthermore, in 2016, Toda et al., showed that the mutational status of *KRAS* is not relevant in its glutamine dependence. These authors demonstrated that KRAS could be associated with asparagine metabolism since asparagine synthetase (ASN) is upregulated in KRAS mutated cells through activation of PI3K/AKT/mTOR signalling pathway. Additionally, these authors point out that it is an adaptive mechanism for tolerating glutamine depletion [159]. Thus, knock out of ASNS led to a dramatic drop in tumour growth in vivo. To date, inhibitors of ASNS have been developed, but not tested in CRC. However, given the importance of PI3K/AKT/mTOR pathway, it would be relevant to define the role played by RHEB, whose modulation could be interesting in regulating asparagine metabolism in these cancers.

#### 3.1.6. Novel Alternatives for Ras Targeted Therapies

The failure of many of the therapies discussed above is leading to the search for different alternatives. One of these new approaches is based on synthetic lethality to target cancer cells with mutant *KRAS*. The development of screenings, mostly based on CRISPR/Cas9 technology or small interfering RNAs (siRNAs), to identify targets of synthetic lethality has been growing in recent years [160]. Several candidate target genes have been identified in the context of CRC, and most of them have available inhibitors (Table 3). However, the efficacy of this strategy has recently been called into question because of its dependence on both environmental changes and the genetic background of the cell models used, making them difficult to reproduce [161].

Another alternative method proposed to target RAS is the use of RAS-inhibiting siRNAs delivered in nanoparticles. It has the advantage of being mutant specific, allowing the targeting of virtually all described mutations [172]. AZD4785 was the first successful drug developed with this technology. It achieved a significant reduction of both WT and mutant KRAS, impairing cell growth of *KRAS*-mutated tumour cells [173]. Cell and Patient-Derived Xenografts models demonstrated in vivo the efficacy of this treatment [126]. However, despite it being well tolerated, it failed in the phase 1 trial, probably due to it targeting both WT and mutant KRAS for degradation. Subsequently, a mutant-specific siRNA against G12D KRAS (siG12D-LODER) was designed. It has shown good efficacy and tolerability in phase 1 trials (https://clinicaltrials.gov/ct2/show/NCT01188785 accessed on 10 May 2021), and it is currently under clinical evaluation in a phase 2 trial to assess the response rate when it is combined with chemotherapy treatment in pancreatic cancer patients with *KRAS* mutations (https://clinicaltrials.gov/ct2/show/NCT01676259 accessed on 10 May 2021).

Autophagy has aroused as an alternative for targeting Ras signalling [48]. It has been shown that inhibition of KRAS as well as its downstream signalling molecules [174], as RHEB [58], is associated with decreased autophagy. It is known that autophagy makes CRC cells more aggressive and favour adaptation to apoptotic stimuli [175] (discussed in the next section). Therefore, it is plausible that inhibition of autophagy would be effective for CRC treatment in mutant KRAS patients. Hydroxychloroquine, an FDA-approved drug for treating malaria, is an inhibitor of autophagy that has been used in preclinical research and clinical trials to test this hypothesis in the context of cancer. Hydroxychloroquine leads to a drop in ROS production in immune cells, increasing immune response and leading to cell cycle arrest in tumour cells [175]. Unfortunately, it showed limited activity when administered as monotherapy in phase 1 clinical trials, but it is getting better results in phase 2 trials in combination with FOLFOX and Bevacizumab, increasing the overall survival from 68% to 74%, and being well tolerated by the patients [176].

Generating oxidative stress is another approach proposed for beating oncogenic *RAS*-mutant cancers. It has been typically assumed that oncogenic RAS enhances pro-oxidative programs that mediates tumorigenesis. However, during last years it has been demonstrated that it also mediates anti-oxidative responses, most of them mediated by the upregulation of NRF2 (Nuclear Factor Erythroid 2-Related Factor 2), one of the main protectors against oxidative damage (reviewed by Jaganjac et al. 2020) [177]. NRF2 inhibition has been shown to impair cell growth and improve the response to chemotherapy in pancreatic [153] and lung cancers [178]. Indeed, it has been shown that NRF2 regulates the expression of multidrug resistance (MDR) proteins in response to oxidative stress [179], which are major mediators of chemotherapy resistance by drug efflux outside the cells in CRC [180,181]. RHEB [182] and RALA [183] dysregulation have also been linked to ROS (reactive oxygen expression) production and oxidative stress. Regarding that PI3K/AKT/mTOR signalling is positively associated with MDR expression [184], it would be interesting to investigate not only the role of KRAS and other Ras-GTPases in the development of resistance to chemotherapy through anti-oxidative responses.

However, the best-described protector against Reactive Oxygen Species (ROS) in CRC is TAK1 (Transforming Growth Factor β-activated kinase 1). It has been identified as a pro-survival signalling pathway in *KRAS* mutant cell lines and its loss increases ROS, leading to cell death. Inhibition of thioredoxin reductase enhances the effects of TAK1 inhibition [167]. Regarding that thioredoxin oxidation is implicated in reducing oxidative stress, it highlights the potential of this anti-oxidative approach for treating *KRAS*-mutated CRC.

Finally, ferroptosis is a novel process related to cancer and, especially, to those harbouring *KRAS* mutations. Ferroptosis is a form of apoptosis independent of caspase in which cells undergo death through iron-dependent lipid peroxidation [185]. It has been suggested that cells evading other mechanisms of cell death are more sensitive to ferroptosis [186], justifying its therapeutic consideration. In CRC, it has been shown that Bromelain, a mixture of proteolytic enzymes used to treat other pathologies, simulates ferroptosis in *KRAS*-mutated CRC, being cytotoxic for *KRAS*-mutated cells but not for *KRAS*-WT [187]. These results are in accordance with those obtained in other types of cancer, thus opening new opportunities for developing new strategies for the treatment of *KRAS*-mutated tumours.

### 3.2. Anti-Ras Strategies: A New Ally for Improving Current EFGR-Targeted Therapies?

Anti-EGFR therapies were a step forward in treating mCRC patients. However, deregulation in EGFR downstream elements leads to both primary and acquired resistance by accumulating genetic alterations or clonal selection of subpopulations that previously harbour these mutations. The role of the Ras-GTPases subfamily in cetuximab resistance is well known and the presence of activating mutations in its members and their downstream signalling molecules are especially important, but not the only ones responsible for these phenomena. Indeed, the analysis of *RAS* mutational status in primary and acquired resistant patients is a good example of these different dynamics. While mutations in exon 61 of *KRAS* and *NRAS* are very infrequent at diagnosis, they are overrepresented in patients who have developed acquired resistance. Given that all oncogenic RAS isoforms are associated to a greater or lesser extent with resistance to anti-EGFR therapies, targeted therapies against RAS could provide an alternative approach to increase sensitivity to these treatments.

Following the order described in the previous section, the first strategy to reverse resistance to anti-EGFR therapies in *KRAS*-mutated patients is to corrupt the subcellular localisation of Ras-GTPases [70]. Evidence for this possibility comes from statins, HMG-CoA reductase inhibitors, which inhibit the mevalonate pathway by directly interfering with farnesylation processes [188]. In vitro trials using CRC cells did show a synergistic effect when statins and EGFR monoclonal antibody were combined [189], but their use in clinical trials did not restore the sensitivity to cetuximab in patients harbouring *RAS* mutations [190].

Likewise, drugs targeting specific KRAS mutants, mostly G12D, have emerged as an, a priori, good strategy in combination with anti-EGFR therapies. Several studies have demonstrated the efficacy of co-targeting *KRAS* G12C and EGFR. For example, McFall and colleagues have studied the role of Sotorasib and Cetuximab combination in CRC cell lines in which various *KRAS* mutations were generated. The results showed a synergistic effect of both drugs in cells with G12C mutation, but not in those with other types of mutations [191]. Patricelli et al., obtained similar results using lung cancer cell lines and combining Erlotinib and ARS-853, EGFR and KRAS G12C inhibitors, respectively [192].

Furthermore, it has been shown that in cancer cells with stem-like properties, RALB is recruited by the αvβ3 integrin to the plasma membrane together with KRAS and NF-κβ. This leads to tumour initiation, anchorage-independent cell growth and resistance to EGFR inhibitors such as Erlotinib in pancreatic, breast and lung cancer. The αvβ3 integrin play a relevant role in CRC, mediating metastasis and angiogenesis processes [193], and is associated with patient survival [194]. Inhibition of this pathway has been shown to be sufficient to reverse resistance to Erlotinib in different tumours [195], so the use of specific inhibitors against RALB could be a good strategy in CRC to reverse resistance to EGFR-targeted therapy, especially in those cases in which KRAS inhibition is not possible. Thus, the design of in vitro and in vivo models to elucidate the role of RALB, together with KRAS, in the development of resistance to anti-EGFR therapies in CRC is necessary.

Inhibition of KRAS downstream signalling molecules has been an alternative approach to reverse resistance to anti-EGFR therapies. Several preclinical models in CRC have shown the success of combinatory therapies in inhibiting BRAF and EGFR simultaneously. One of these studies, conducted in xenograft models with *BRAF*-mutant tumours, showed that treatment with vemurafenib (a specific BRAF inhibitor) was enough to reverse resistance to EGFR inhibitors [135]. As discussed, *BRAF* mutations are generally mutually exclusive with *RAS* mutations and this strategy would not be effective in most of the RAS mutant tumours. However, given that BRAF is immediately downstream of RAS, its involvement in resistance to anti-EGFR therapies is evident [196]. Thus, the inhibition of BRAF could be a good approach to reverse resistance in patients with *RAS* mutations. In this regard, Martinelli and co-workers conducted an in vitro trial of combinatorial therapies with Sorafenib, a non-specific RAF inhibitor, and different anti-EGFR therapies in *KRAS*-mutated cell lines. The results showed a strong ability of Sorafenib to reverse resistance to both Cetuximab and Erlotinib [197]. However, randomized clinical trials of metastatic CRC evaluating Sorafenib in combination with cetuximab improved the partial response to cetuximab only in *KRAS*-WT patients but not in those harbouring *KRAS* mutations [198]. Additionally, the inhibition of MEK1/2, located downstream from BRAF, with BAY 86-9766 achieved a reversion of cetuximab resistance in cell-line derived xenograft models [199]. This drug was well tolerated in phase 1 clinical trial, but it has not been tested in combination with anti-EGFR therapies yet.

Targeting metabolism has emerged as a good strategy for managing *KRAS*-mutated CRC and, therefore, for overcoming anti-EGFR resistance. Indeed, it has been demonstrated in preclinical trials that Vitamin C is enough to sensitize *KRAS*-mutant cells to cetuximab in CRC [151] and Erlotinib in lung cancer [200], both in vivo and in vitro. Unfortunately, although Vitamin C is under clinical trials in combination with the anti-EGFR panitumumab in WT *KRAS* patients, no clinical trials targeting both metabolism and EGFR have been performed for *KRAS*-mutated CRC patients so far. Regarding amino acid metabolism, it has been related to drug resistance in PDAC [153]. However, available clinical trials combining CB-839 glutaminase inhibitor with panitumumab only recruit CRC patients with WT *KRAS*. 

Autophagy is emerging as an indirect pathway to inhibit constitutively active RAS and reverse anti-EGFR resistance. Cetuximab-mediated apoptosis is generated, at least in part, by an increase in autophagy levels generated by inhibition of the PI3K/AKT pathway [201]. In this regard, it has been shown that overactivation of this pathway decreases autophagy levels and favours resistance phenotypes. Combined inhibition of EGFR and mTOR leads to resensitisation through the activation of autophagy [202]. Currently, there are FDA-approved inhibitors against mTOR pathway for other purposes, it is necessary to explore their efficacy to reverse anti-EGFR resistance in these tumours and looking for markers to elucidate which patients could benefit from this strategy.

Finally, ferroptosis is being related in recent years to drug resistance [203], and this approach has been applied to overcome anti-EGFR resistance. Chen et al. demonstrated that β-elemene, combined with cetuximab, was enough to reduce tumour growth both in vitro and in vivo. This compound reversed cetuximab resistance by inducing ferroptosis and inhibiting epithelial-to-mesenchymal transition [203]. However, ferroptosis modulation is not under clinical trials in combination to anti-EGFR agents.

## 4. Conclusions

Recent advances in RAS-targeted therapies have brought this protein back to the centre of the target as alternative treatments in CRC with oncogenic RAS. Direct inhibitors for specific KRAS mutants, mainly G12C, seem to be just the beginning of a new generation of drugs that improve the prognosis of these patients. Likewise, evidence for the involvement of non-classical Ras-GTPases in CRC is growing. Many of these Ras-GTPases, such as RALA, RALB and RHEB, also act as mediators of oncogenic RAS signalling and have an increasing number of specific inhibitors. Therefore, investigating the role of these Ras-GTPases is important for the development of new therapies for the treatment of CRC patients.

The increasing knowledge about new cellular processes associated with *KRAS* mutations in CRC is providing new potential targets to treat these tumours. Therefore, reprogramming metabolism, responses to oxidative stress, and ferroptosis need to be deeper studied to develop inhibitors that could be clinically useful for these patients. Although anti-EGFR therapies have significantly improved patient survival, the number of responders is limited. Both RAS and its downstream signalling molecules are main characters in the development of resistance to these therapies, so the combination of inhibitors targeting these proteins could be interesting not only to sensitize patients to anti-EGFR therapies but also to slow down the appearance of resistance associated with RAS inhibitors.

Therefore, over the next few years, it will be necessary to maintain efforts to refine these combinatorial therapies and their evaluation in both preclinical and clinical trials. The success of this approach could greatly increase the number of patients eligible for treatment with anti-EGFR therapies.

## Figures and Tables

**Figure 1 cancers-13-03757-f001:**
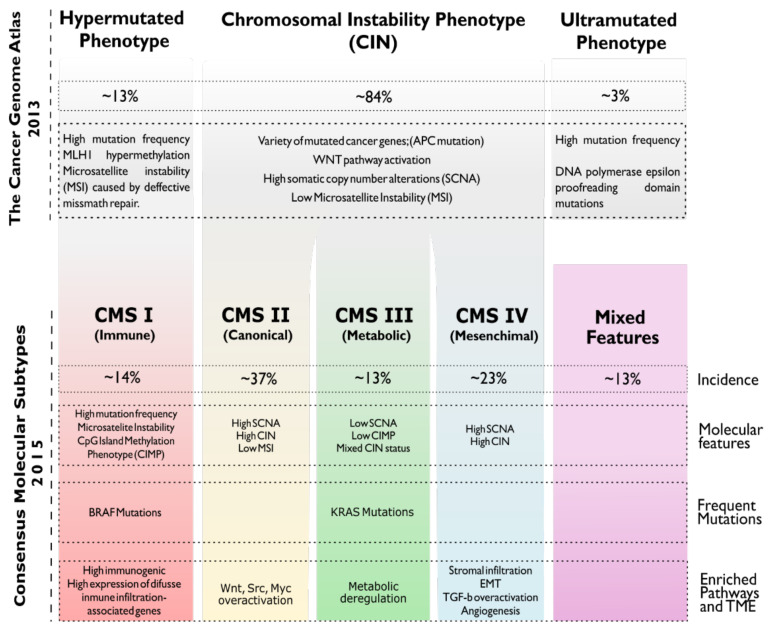
Molecular classifications of CRC tumours. The Cancer Genome Atlas Classification (TCGA) divides CRC tumours into 3 groups according to their mutational status. Meanwhile, Consensus Molecular Subtypes Classification (CMS) proposes to establish 4 groups according to microsatellite instability status (MSI), chromosomal instability (CIN), methylation patterns (CIMP) as well as the most frequently altered pathways, *KRAS* and *BRAF* mutational status and tumour microenvironment status (TME).

**Figure 2 cancers-13-03757-f002:**
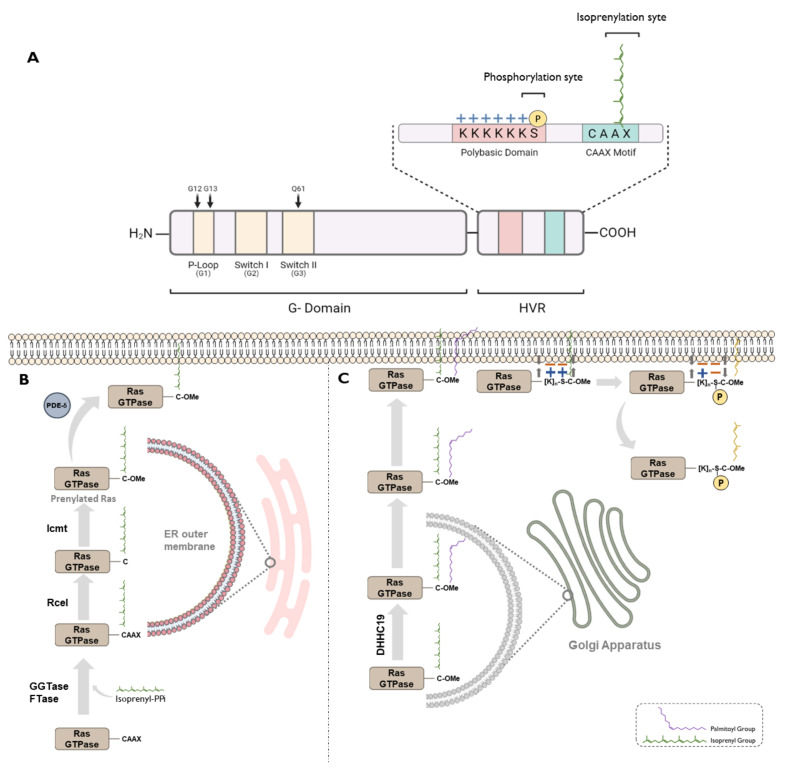
(**A**) General protein structure of Ras-GTPase members. (**B**) Regulation of Ras-GTPase membrane location by isoprenylation (farnesylation and geranylgeranylation). (**C**) Regulation of Ras-GTPase membrane location by palmitoylation and phosphorylation.

**Figure 3 cancers-13-03757-f003:**
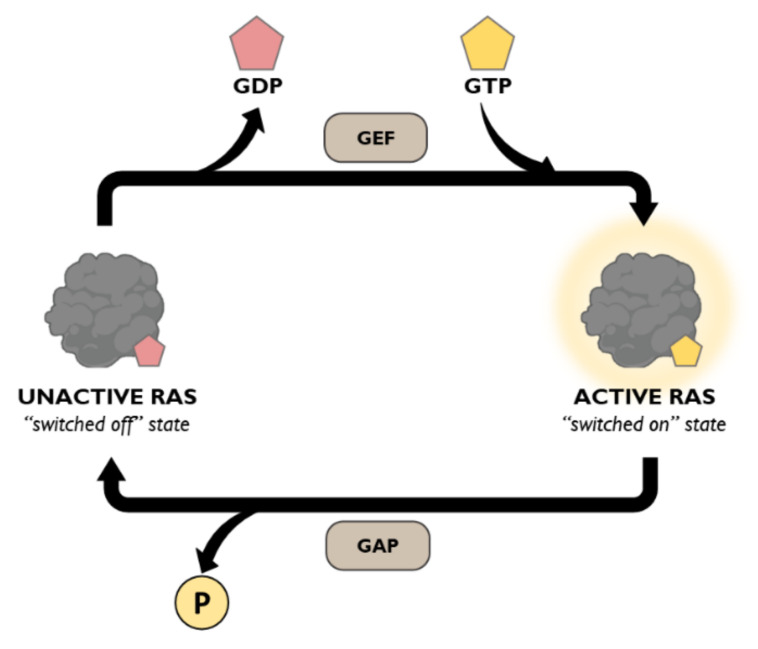
Diagram of Ras GAP/GEF Cycle. GAP proteins hydrolyse GTP to GDP, promoting Ras to reach an inactive state, while GEF proteins mediate the GDP-to-GTP exchange leading to Ras active conformation.

**Figure 4 cancers-13-03757-f004:**
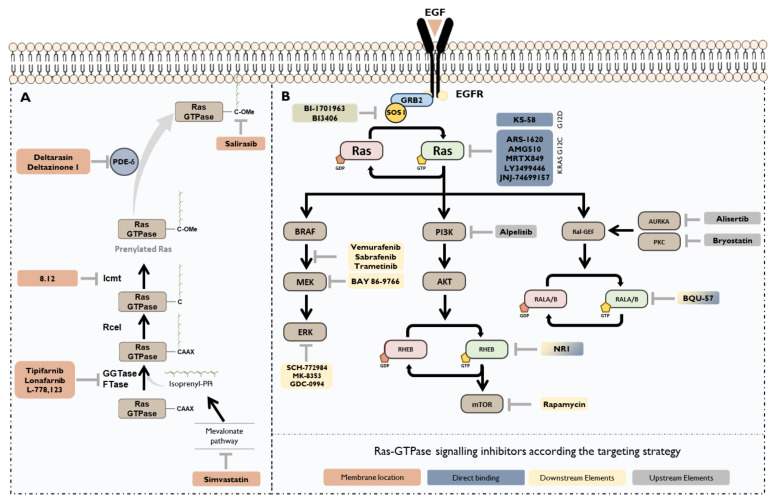
Current strategies for targeting RAS-GTPases in CRC. (**A**) Targeting Ras membrane location. Inhibition of Ras prenylation or its coupling to plasma membrane has been used both in preclinical and clinical trials for treating KRAS mutated tumours. (**B**) Targeting Ras family members directly or its upstream or downstream elements.

**Table 1 cancers-13-03757-t001:** Members of Ras-GTPase family, functions, and its implications in cancer. Ras-GTPase family is formed by 6 subfamilies (RGK, RAL, RAP, RAS, RHEB and RIT), most of them implicated on development and progression of several types of cancer.

Ras Subfamilies	Main Functions	Members	Implications in Cancer	Refs
RGK	Cytoskeleton remodelling Voltage-dependent Ca-Channels Insulin-induced glucose uptake	REM1, REM2, RAD, GEM	No consistent evidence of its role in cancer	[33,34]
RAL	Mediator of exocytosis Cell proliferation Cell migration	RALA RALB	Mediates Ras oncogenic transformation. Implication in colorectal, pancreatic, bladder, lung, prostate cancer, and melanoma.	[35,36,37,38,39]
RAP	Integrin-mediated cell adhesion Formation of cell–cell junctions Establishment of cell polarity Exocytosis Apoptosis Cell proliferation	RAP1A, RAP1B, RAP2A, RAP2B, RAP2C	Implication in squamous cell and head and neck carcinoma, breast, colorectal, brain, lung	[40,41,42,43,44,45,46,47]
RAS	Cell proliferation Cell adhesion Cell differentiation Organ development Neuronal plasticity	ERAS, NRAS, HRAS, KRAS, MRAS, RRAS RRAS2, DIRAS, DIRAS2, DIRAS3 NKIRAS1 NKIRAS2 RASD1 RASD2 RASL10A RASL10B RASL11A RASL11B RASL12 RERG	Oncogenic branch of Ras-GTPases (KRAS, HRAS, NRAS) are implicated in almost all types of cancer. DIRAS subgroup counteracts oncogenic Ras acting as a tumour suppressor in breast and ovarian cancer. RASD1, RASL11A and RERG seems to act as tumour suppressors in glioblastoma, prostate cancer, and nasopharyngeal carcinoma, respectively.	[32,48,49,50,51,52,53]
RHEB	Cell growth Cell cycle control Autophagy Amino acid uptake	RHEB, RHEBL1	Related with metastasis in prostate cancer and increasing survival and proliferation in CRC cell lines.	[54,55,56,57,58]
RIT	Neuronal differentiation and survival	RIT1, RIT2, RIN, RIC	Not described	[59]

**Table 2 cancers-13-03757-t002:** Clinical trials testing Ras-GTPase inhibitors in CRC patients with *KRAS* or *BRAF* mutations.

Biomarker	Inhibitor	Strategy	Phase	Status	Study Identifier
PKC	Bryostatin	Targeting Location	II	Completed	NCT00003220
RAS Farnesyl	L-778 123	Targeting Location	I	Completed	NCT00003430
RAS MBS *	Salirasib	Targeting Location	II	Completed	NCT00531401
KRAS^G12C^	MRTX-849	Direct Targeting	III	Recruiting	NCT04793958
KRAS^G12C^	LY3499446	Direct Targeting	I	Terminated	NCT04165031
KRAS^G12C^	JNJ-74699157	Direct Targeting	I	Completed	NCT04006301
SOS1	BI-1701963	Targeting Upstream Elements	I	Recruiting	NCT04111458
MEK	Trametinib	Targeting Downstream Elements	I	Recruiting	NCT03714958
ERK	MK-8353	Targeting Downstream Elements	I	Active	NCT02972034
-	Vitamin C	Targeting Metabolism	II	Recruiting	NCT03146962
-	Vitamin C + Carbohydrate restriction	Targeting Metabolism	I	Not yet recruiting	NCT04035096

* RasMBS: Ras membrane binding sites.

**Table 3 cancers-13-03757-t003:** RAS synthetic lethal genes. Screenings for searching targets for RAS synthetic lethality in CRC. Many of them were accompanied by drug-based inhibition assays.

Genes	Type of Screening	Cell Lines Used	Drug Inhibition	Refs
PLK1	Genome scale shRNA screening	DLD1 (KRAS G13D)	BI-2536	[162,163]
Survivin	siRNA screening of ~4000 genes	DLD1 (KRAS G13D)	Not tested	[164]
SNAIL2	shRNA screening of ~2500 cancer-related genes	HCT116 (KRAS G13D)	Not tested	[165]
GATA2 CDC16	siRNA screening of ~7000 genes	HCT116 (KRAS G13D)	Bortezomib (proteosome inhibitor) + GATA2 silencing	[166]
TAK1	Screening of 17 kinases, using 5 shRNAs	SW620 (KRAS G12V) SW837 (KRAS G12C) HCT116 (WT) HCT116 (KRAS G13D)	NG25 (in vitro and in vivo)	[167,168]
BCLXL MEK	Screening of genes whose inhibition cooperate with MEK inhibitors	SW620 (KRAS G12V) HCT116 (WT) HCT116 (KRAS G13D)	Selumetinib and navitoclax (MEK inhibitors)	[169]
CDK1	siRNA library targeting 784 genes.	Isogenic LIM1215 (KRAS WT and mutant)	RO-3306 (in vitro and in vivo)	[170]
RAF1	shRNA library targeting 535 kinases and related genes	SW480 (KRAS G12V)	RAF265 or AZ628 (RAF inhibitors) with selumetinib	[171]

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
