# Peer review of "Ras Family of Small GTPases in CRC: New Perspectives for Overcoming Drug Resistance"

_cancers, 2021, doi:10.3390/cancers13153757_

Round 1

Reviewer 1 Report

A timely review article by Dr. Cebrián and group, elaborating on the involvement of mutant RAS signaling in creating resistance in CRC patients, and also it has been discussed the therapeutic opportunities to combat resistance. Few things need to be addressed before it is ready for acceptance. They are as follows:

  1. It has been recently discussed how RAS plays a broader role in drug resistance (doi: 10.1038/s43018-021-00184-x), authors need to include this and discuss few lines on this perspective. Definitely, metabolism plays a significant role in RAS-driven drug resistance. The authors should add few lines discussing this. 
  2. Moreover, it has been shown in 10.1158/0008-5472.CAN-19-1363, how targeting metabolism would be beneficial to combat drug resistance in RAS-driven cancers. For therapeutic purposes, targeting glutamine pathways or sometimes glucose pathway is beneficial to combat resistance in RAS-mutant cancers. Authors must add few lines while discussing the possible therapeutic opportunities to combat resistance.
  3. Authors should add a table mentioning all clinical trials occurring in RAS-driven CRC, this will give readers some update on RAS-driven cancer's therapeutics from a patient perspective.
  4. Antioxidant pathways play a definite role in drug resistance (doi: 10.1155/2016/4251912 and 10.1158/0008-5472.CAN-19-1363). The authors should add few lines discussing that perspective on basis of RAS-driven CRC.
  5. Also, recently ferroptosis (DOI: 10.1038/s41568-019-0149-1)has been introduced as another possible pathway that is involved in drug resistance. It will be worthwhile if the authors shed some light on this in the discussion part as a future path to study further. 

Reviewer 2 Report

In this review, Rio-Vilariño et al revised the current available information about the role of Ras-GTPases in the initiation and progression of colorectal cancer and their relationship with resistance to targeted therapies, as well as the status and new advances in the design of targeted therapies against these proteins and their possible clinical implications.

The review is very well written, with well-structured and developed subsections as well as extensive updated bibliography. The figures are adequately illustrated.

I recommend publication of this review in Cancers if minor comments are addressed:

1) The authors indistinctly use the term RAS GTPases, Ras-GTPases, Ras- GTPases or Ras GTPases. Please homogenize.

2) In line 18 “….the EGFR and VEGF membrane receptors..”. Please correct.

3) In line 46, what does APC mean? Please define.

4) The authors indistinctly use the term 5FU and 5-FU. Please homogenize.

5) The authors indistinctly use the term MAPK and MAP-Kinase. Please homogenize.

6) In line 236, please confirm that affected codons are 12 and 14 in KRAS and HRAS instead 12 and 13.

7) The authors sometimes use indistinctly British and American English styles (i.e signaling and signalling). Please homogenize.

8) In section 3.1.2, when describing AMG-510, it would be useful to define that this molecule has been named as Sotorasib (I recommend to use this terminology instead AMG-510). In addition, results about Phase II clinical trial (NCT03600883) are already available and published. Moreover, a phase III trial already started (NCT04303780). This information should be included in this subsection.

9) Although slightly mentioned, when the authors describe strategies to inhibit Ras upstream effectors in section 3.1.3 (also in Figure 4), they should clearly indicate that BI-1701963 and BI-3406 specifically affects Sos1, but not Sos2, interaction with KRAS. In this regard, the author could consider to include in this description the molecule BAY-293.

Round 2

Reviewer 1 Report

All concerns have been addressed, ready for acceptance.